# Characterization of Polylactic Acid Membranes for Local Release of Tramadol

**DOI:** 10.3390/ijms26136018

**Published:** 2025-06-23

**Authors:** Lafitte Fernández-Minotre, Mauricio Montero-Aguilar, Febe Carolina Vázquez-Vázquez, Janeth Serrano-Bello, José Vega-Baudrit, Reinaldo Pereira-Reyes, Amaury Pozos-Guillén, Daniel Chavarría-Bolaños

**Affiliations:** 1Dentistry Graduate Program, Universidad de Costa Rica, San José 11501-2060, Costa Rica; lafitte86@hotmail.com (L.F.-M.); mauricio.monteroaguilar@ucr.ac.cr (M.M.-A.); 2Laboratorio de Materiales Dentales, DEPeI-Facultad de Odontología, UNAM, Ciudad Universitaria, Coyoacán, Ciudad de México 04510, Mexico; fcarolina.vazquez@gmail.com; 3Laboratorio de Bioingeniería de Tejidos, DEPeI-Facultad de Odontología, UNAM, Ciudad Universitaria, Coyoacán, Ciudad de México 04510, Coyoacán 04510, Mexico; janserbello@fo.odonto.unam.mx; 4Laboratorio Nacional de Nanotecnología, Centro Nacional de Alta Tecnología, San José 10109, Costa Rica; jvegab@gmail.com (J.V.-B.); rpereira@cenat.ac.cr (R.P.-R.); 5Basic Science Laboratory, Facultad de Estomatología, Universidad Autónoma de San Luis Potosí, San Luis Potosi 78290, Mexico; apozos@uaslp.mx

**Keywords:** scaffold, surgical membrane, tramadol, polylactic acid, air jet spinning, drug delivery system

## Abstract

This study aimed to develop polylactic acid (PLA)-based membranes incorporating tramadol (TMD) using air jet spinning (AJS), ensuring stable physicochemical properties and biocompatibility. Two groups were fabricated: 10% PLA membranes (control) and 10% PLA membranes loaded with TMD in an 80:1 ratio (experimental). Characterization included scanning electron microscopy (SEM), differential scanning calorimetry (DSC), thermogravimetric analysis (TGA), Fourier-transform infrared spectroscopy (FT-IR), ultraviolet-visible spectroscopy (UV-VIS), and biocompatibility assays with human osteoblasts using resazurin, crystal violet staining, and 5-chloromethylfluorescein diacetate for fluorescence microscopy. SEM revealed a homogeneous, randomly distributed fiber pattern, with diameters under 5 µm and no structural voids. DSC and TGA indicated that TMD was uniformly incorporated, increased the thermal capacity, and slightly lowered the onset and inflection degradation temperatures. FT-IR confirmed the chemical compatibility of TMD with PLA, showing no structural alterations. UV-VIS detected sustained TMD release over 72 h. Biocompatibility tests showed no cytotoxic effects; cell viability and proliferation in TMD-loaded membranes were comparable to controls. Statistical analysis used ANOVA and Wilcoxon tests. 10% PLA membranes loaded with TMD at an 80:1 ratio exhibited stable physicochemical characteristics and favorable biocompatibility, supporting their potential use in drug delivery systems.

## 1. Introduction

Postoperative pain is one of the main concerns dentists must address when performing surgical procedures such as bone regeneration or augmentation. Pain is an unpleasant, multidimensional sensation that varies in intensity, quality, duration, and location. It is classified into several types: nociceptive, inflammatory, neuropathic, and functional [1]. Tissue engineering procedures for bone regeneration inevitably produce tissue damage. Due to local anesthesia, the nociceptive system initially ignores the injury, delaying the body’s defensive mechanisms. This leads to a systemic response involving inflammatory pain, which promotes healing by increasing sensitivity at the surgical site. This heightened sensitivity helps prevent further tissue damage by reducing movement or friction at the healing site; even standard stimuli that typically do not cause pain may provoke a painful response in these hypersensitive areas. Therefore, clinicians must pharmacologically manage inflammatory pain without interfering with the healing process. The goal of anti-inflammatory/analgesic treatment should be to modulate rather than completely suppress symptoms [2].

To achieve effective pain control and ensure patient comfort, analgesics are commonly prescribed systemically after the procedure. Tramadol (TMD) is a long-standing analgesic with notable potential and versatility due to its moderate analgesic effect, synergy with anti-inflammatories, low risk of allergic reactions, and efficacy in managing incisional pain. Common side effects of TMD include nausea (6.1%), dizziness (4.6%), drowsiness (2.4%), fatigue (2.3%), sweating (1.9%), vomiting (1.7%), and dry mouth (1.6%). Although TMD has proven to be a broad-spectrum analgesic—either as a single agent or in combination with other drugs—its side effects may prompt clinicians to consider alternative approaches [3]. To reduce these adverse effects, lower doses of TMD administered locally at the injury site have been recommended. Local analgesia achieves high drug concentrations at the target site while maintaining low systemic levels, thereby limiting side effects. This approach also minimizes pharmacological interactions and allows broader use without being constrained by tolerability thresholds [4]. Positive clinical outcomes have been reported when TMD was placed into the tooth socket following third molar extractions, enhancing healing and patient comfort [5]. Additional studies have evaluated the efficacy of combining TMD with local anesthetics such as lidocaine, mepivacaine, and articaine in maxillary infiltration procedures. For example, in combination with lidocaine, TMD enhanced the anesthetic effect, healing, and pain perception [6]. TMD-mepivacaine combinations yielded promising results in inferior alveolar nerve blocks, while TMD-articaine synergy was clinically demonstrated in impacted third molar surgery and the treatment of symptomatic irreversible pulpitis [7,8].

Peripheral analgesia has been achieved through various delivery systems, including topical gels or creams, patches, aerosols, mouthwashes, and membranes for localized drug release [3]. Membranes are widely used in dentistry, particularly for regenerative procedures such as guided bone regeneration (GBR). They are a key component of tissue engineering, which involves three main elements: cells, the extracellular matrix (ECM), and signaling molecules. Controlled manipulation of the extracellular environment aims to promote cellular organization, growth, differentiation, ECM formation, and ultimately, new tissue development [9]. Membranes serve as barriers to prevent soft tissue ingrowth in GBR and as scaffolds that support soft tissue reconstruction during wound healing. Ideal membrane properties include biocompatibility, cell occlusiveness, host tissue integration, ease of clinical handling, space maintenance, and suitable physicomechanical characteristics. Resorbable membranes represent the second generation of barrier materials, developed to avoid the need for a second surgical procedure to remove non-resorbable membranes [10]. These membranes are typically produced using polymers and various fabrication techniques. Polymer-based materials form porous, three-dimensional membranes designed to mimic ECM geometry and support cellular attachment and proliferation [11,12]. Biodegradable polymers used in membrane fabrication are classified as either natural or synthetic. Polylactic acid (PLA) is a widely used synthetic polymer with considerable potential in biomedical applications [9]. PLA is valued for its ease of processing, biodegradability, low cost, and ability to encapsulate drugs [11,13,14,15,16,17].

Among the most common fabrication methods for polymeric membranes are electrospinning and air jet spinning (AJS). AJS offers distinct advantages for industrial scalability: it is faster, easier to implement, more cost-effective, safer (as it does not involve high voltage), and allows for solvent flexibility. AJS employs pressurized gas to stretch the polymer solution into fine fibers, which are deposited onto a substrate, producing micro- and nanoscale fibers from various polymers [10,17,18,19]. Polymer therapy is an evolving field in biopharmaceuticals that utilizes linear or branched polymer chains as bioactive elements covalently linked to therapeutic agents. This strategy improves the pharmacokinetic and pharmacodynamic profiles of drugs by extending plasma half-life (thus requiring less frequent dosing), enhancing resistance to enzymatic degradation, reducing immunogenicity, improving protein stability and solubility, and enabling targeted local action [20,21]. The development of efficient, biocompatible, biodegradable membranes that can support tissue regeneration while providing controlled local drug delivery—specifically of analgesics—remains an ongoing challenge. Therefore, the objective of this study was to fabricate TMD-loaded PLA nanofibers using the AJS technique to produce resorbable membranes with stable physicochemical properties and proven biocompatibility.

## 2. Results

### 2.1. Macro and Microphotograph Analysis

It was possible to synthesize 10% PLA membranes loaded with TMD at an 80:1 polymer-to-drug ratio. Macroscopically, these membranes exhibited a soft texture and a white-translucent appearance. Although they can be isolated, separated, and cut, their handling requires care to avoid distortion of the biomaterial. Under 40× magnification using light microscopy, a random arrangement of PLA fibers was observed in both groups, with regions of varying membrane density. At lower SEM magnification (50×), differences between the groups were not readily distinguishable. The membranes showed a random distribution of polymer fibrils without voids or empty areas. At higher magnifications (2000× and 5000×), longitudinal fibers were observed, oriented in multiple planes and angulations (Figure 1). The average fiber diameter was 700 ± 110 nm in the control membranes and 680 ± 120 nm in the TMD-loaded membranes, with no statistically significant differences between groups (*p* > 0.05) (Figure 2). In some regions, the formation of plates or larger fiber constructs was visible, likely resulting from the precipitation and aggregation of individual fibers.

### 2.2. Analysis of Thermodynamic Profile

#### 2.2.1. Differential Scanning Calorimetry (DSC)

It was observed that the control group exhibited different calorimetric curves compared to the pure polymer (PLA pellet), which did not present the first endothermic signal and displayed a well-defined second signal (Figure 3). In the experimental group, calorimetric analysis revealed the absence of the endothermic peak corresponding to pure TMD, suggesting that the drug was homogeneously and stably incorporated into the polymeric system. DSC analysis was subsequently performed five times for both the control and experimental groups to assess sample purity. The maximum temperature and heat capacity of the first thermal event showed no statistically significant differences between groups (*p* > 0.05). However, a statistically significant increase in heat capacity was observed in the TMD-loaded membrane (*p* < 0.05), indicating that the presence of the drug elevates the heat absorption capacity of the polymeric system (Figure 4).

#### 2.2.2. Thermogravimetric Analysis (TGA)

TGA confirmed the absence of impurities or by-products during the membrane fabrication process. Both the 10% PLA membranes and the 10% PLA-TMD membranes exhibited a single mass loss event, characterized by a clear onset temperature (To) and a single inflection point (Tp), calculated from the derivative of the weight percent change with respect to temperature (Table 1, Figure 5). Similarly, pure tramadol displayed a single-step degradation curve with defined To and Tp values (Table 1).

### 2.3. Chemical Composition

#### 2.3.1. FT-IR Analysis

FT-IR analysis demonstrated that the manipulation of the polymer during membrane synthesis did not alter its chemical composition. Likewise, the incorporation of TMD was compatible with the polymer and did not significantly modify the infrared spectral profile. Specific signal changes at 2900, 1650, and 1500 cm^−1^ were observed in the TMD-loaded membranes but not in the control group; these were attributable to the presence of tramadol. At higher magnification, a symmetric CH_2_ stretching signal was identified between 2900 and 2800 cm^−1^, along with the asymmetric methylene group signal at 2933 cm^−1^, both previously described for this molecule. Other characteristic tramadol peaks may have remained undetected due to the low drug concentration and its incorporation into the polymer matrix prior to AJS synthesis (Figure 6).

#### 2.3.2. UV-VIS Spectrometric Analysis

The presence of TMD was confirmed in the recovered aliquots, with concentrations of 185.31 ± 2.10 µg/mL, 180.80 ± 15.27 µg/mL, and 174.88 ± 16.00 µg/mL at 24, 48, and 72 h, respectively. These findings demonstrate that TMD was successfully loaded into the membranes and could be released and recovered through hydrolysis of the system. In contrast, none of the aliquots recovered from the control membranes showed the presence of tramadol or any absorption changes at the corresponding wavelength. Figure 7 illustrates the percentage distribution of recovered tramadol over the first 72 h.

### 2.4. Biocompatibility Analysis

The biocompatibility of PLA fiber spun mat and PLA loaded with TMD was analyzed in vitro in cell culture to investigate the cell viability of hFOB 1.19 cells. The results are presented as the optical absorbance at 545 nm. The histogram in Figure 8 suggests that in the scaffolds, the presence of tramadol did not show any adverse effects on cell viability, and there were no significant differences in comparison with the control PLA without loaded TMD.

#### Colonization of PLA Spun Membrane

The colonization over the surface of the PLA fiber spun mat showed that the presence of tramadol does not affect the interaction of the cells in the topographical fiber cues of the polymeric material. This behavior is in agreement with the viability test, and the green CFMDA also showed that hFOB cells are viable and spreading on the surface of the PLA membrane after 48 h in both the control (Figure 9a) and experimental group (Figure 9c). In addition, the homogeneous distribution of the cells covering the fiber topography of the PLA membranes was observed by SEM micrographs. Also, it appreciated the interaction with the neighboring cells and followed the random orientation of the fibers (Figure 9b,d).

## 3. Discussion

The AJS technique was chosen primarily due to its feasibility in customizing and synthesizing PLA nanofibrillar membranes for drug delivery applications. This method offers key advantages such as low-cost equipment, rapid fabrication, and the absence of a high-voltage source requirement [18,22]. In this study, AJS enabled the reproducible synthesis of 10% PLA membranes, with and without TMD, at a polymer-to-drug ratio of 80:1. As reported by Medeiros et al., who compared the efficiency of AJS and electrospinning, AJS can generate micro- and nanofibers comparable in size to those produced by electrospinning but with greater potential for commercial scale-up. Moreover, it allows coating of various substrates, including materials sensitive to thermal changes or voltage [23].

To ensure optimal membrane formation via AJS, a proper balance between gas flow pressure, solution feed rate, and working distance must be achieved to enable solvent evaporation and fiber solidification. Previous studies have shown that polymer jet behavior—including straight segment length, velocity fluctuations, flapping motion, and evaporation dynamics—affect final fiber morphology. Higher air pressure yields more uniform fibers with smaller diameters, although excessive pressure can result in irregular fibers and strand formation [24].

Stojanovska et al. outlined key parameters for AJS-based membrane and scaffold fabrication, emphasizing polymer concentration, air pressure, nozzle diameter, and synthesis distance [18]. In this study, an air pressure of 30 psi, a 0.3 mm nozzle, and a working distance of 11 cm were selected, based on the methodology of Suárez-Franco et al., who demonstrated improved cell adhesion and proliferation using PLA at 10% concentration versus lower concentrations [12]. While lower PLA concentrations were initially considered, pilot tests showed enhanced fiber formation at 10% when the drug was incorporated. For TMD loading, the 80:1 ratio provided optimal fiber and drug stability. This combination yielded membranes with similar microtopographic features, showing randomly oriented polymer nanofibers in multiple planes and angles, with diameters not exceeding 5 µm. PLA is an excellent biomaterial for scaffolds supporting tissue regeneration. However, its use in bone regeneration has been questioned, mainly due to its mechanical properties, with criticisms often based on membranes fabricated by electrospinning rather than AJS. In contrast, AJS enables better fiber morphology, pore structure, and random deposition, improving physicomechanical characteristics. Nevertheless, the present study noted difficulties in membrane handling. Future work should address this limitation, as suggested by Bharadwaz et al., who improved PLA scaffold strength by blending it with other polymers or modifying its structure [25,26,27].

Thermal analyses (DSC and TGA) assessed the system’s stability, homogeneity, and purity. DSC revealed the absence of the characteristic endothermic peak of pure TMD (typically around 180 °C), likely due to loss of the drug’s crystalline structure after incorporation into the polymer [28]. A change in the heat capacity of the TMD-loaded membrane suggested homogeneous drug dispersion. Similar results were reported by Kumar et al. using DSC to analyze TMD-loaded alginate–gelatin gels, where drug encapsulation formed a polymer–drug inclusion complex [29]. In our study, TMD altered the thermodynamic profile of the membrane, consistent with prior reports that associate mass changes with chemical alterations [30,31]. TGA showed that TMD decreased both To and Tp in 10% PLA membranes, a pattern similar to that reported by Maubrouk et al. in TMD-loaded poly(ε-caprolactone) [32]. Additionally, the absence of impurities in TGA confirmed the integrity of the manufacturing process.

FT-IR analysis confirmed that polymer manipulation for membrane synthesis did not alter PLA composition. Similarly, TMD incorporation was compatible with the polymer matrix and did not disrupt the infrared spectrum, consistent with earlier studies on drug-loaded polymers [28,29,32,33,34,35]. To verify TMD loading and release from the membranes, UV-VIS spectroscopy was used. According to Küçük and Kadıoğlu, TMD can be detected at 271 nm in methanol or water, while PLA exhibits a peak near 230 nm [36]. The controlled release assay confirmed TMD recovery at 24, 48, and 72 h. These findings are promising for clinical use, particularly in managing acute post-surgical pain, as drug release was evident as early as 24 h and maintained a decreasing yet continuous release profile over time. Future studies will explore the correlation between theoretical and actual drug loading, as well as detailed release kinetics. A limitation of this work was the use of only distilled grade 3 water (pH 5.0–7.5 at 25 °C) in the release assay. Subsequent investigations should evaluate membrane behavior in different media, such as varied pH levels or biological fluids (e.g., saliva or blood).

To assess biocompatibility, cytotoxicity tests were performed on both unloaded and TMD-loaded 10% PLA membranes using staining assays for cell adhesion and proliferation. It is well known that cellular behavior is influenced by the surrounding microenvironment; thus, scaffolds must emulate ECM architecture through nanopores and fiber morphology suited to cellular attachment and migration [37,38]. PLA scaffolds are widely accepted as biocompatible. Granados-Hernández et al. showed that 10% PLA membranes synthesized via AJS supported mesenchymal stromal cells without inducing cytotoxicity [22]. In the present study, fetal human osteoblasts stained with crystal violet confirmed the biocompatibility of both membrane types. As the dye binds to chromatin, only viable cells are stained. The results were consistent with previous findings, demonstrating that the membranes promoted cell proliferation [39,40]. While proliferation was slightly reduced in TMD-loaded membranes, it was not inhibited, suggesting that TMD incorporation did not induce acute cytotoxicity.

Cell adhesion was assessed at 24 and 48 h using resazurin staining, a non-toxic assay sensitive to mitochondrial activity and metabolic function [41]. TMD-loaded membranes maintained comparable adhesion to controls across both time points. Cell–membrane interaction was further evaluated by fluorescence microscopy and SEM. Fluorescence images confirmed cell viability, distribution, and morphology. CellTracker™ staining revealed the presence of viable osteoblasts with intact membranes, cytoplasmic extensions, and characteristic hexagonal morphology, suggesting early cell–cell interactions. SEM provided complementary three-dimensional images, confirming membrane integrity and cell morphology. These findings indicate that the TMD-loaded nanofibrous membranes exhibit favorable biological properties for tissue regeneration [38,39]. The low TMD dosage and absence of chemical residues post-synthesis were critical to achieving biocompatibility. As noted by dos Santos, higher TMD concentrations increase in vitro cytotoxicity, although this can be mitigated by the carrier system [42]. Thus, PLA may have attenuated TMD’s cytotoxic potential. This could explain the slightly reduced, yet still positive, proliferation observed in the experimental group compared to the control.

Some limitations of this study should be considered for future experiments. The TMD concentration was selected based on the highest achievable loading without compromising fiber structure. However, in vivo testing is required to evaluate clinical efficacy. Additionally, as previously mentioned, release assays were performed in ideal aqueous conditions; further testing in physiological or pathological fluids under various pH levels is necessary to fully assess membrane behavior in clinically relevant environments.

## 4. Materials and Methods

### 4.1. Polymer Fiber Fabrication of PLA Loaded with Tramadol

Fibrous spun scaffolds were fabricated via the air jet spinning process from PLA polymeric solutions of 10% wt. First, polymeric solutions of 10% wt of PLA were prepared: PLA pellets (C_3_H_6_O_3_; molecular weight (MW) 192,000, Ingeo Biopolymer 2003D Promaplast, Ciudad de Mexico, Mexico) were dissolved in chloroform (CHCl_3_) and stirred for 20 h. After that, anhydrous absolute ethyl alcohol (CH_3_CH_2_OH) was added, and the solution was stirred for 30 min to obtain a homogeneous solution. The volume ratio of chloroform/ethanol was 3:1. Then, the polymeric solution was prepared with tramadol (Tramadol HCL, code 15101782, lot 3735ID12) to obtain a mixture with a volumetric ratio of 80:1 for the composite fiber scaffold. For the synthesis of the fibrillar membrane scaffolds, in all cases, the polymeric solution was placed in a commercially available airbrush, (ADIR model 699) with a 0.3 mm nozzle diameter and with a gravitational feed. The airbrush was connected to a pressurized argon tank (CAS number 7740-37, concentration > 99%, PRAXAIR, Ciudad de Mexico, Mexico). For deposition of the fibers, a pressure of 30 psi with 11 cm of distance from the nozzle to the target was held constant. Once the fibers were obtained, they were subjected to analysis of their physicochemical and structural properties.

### 4.2. PLA Loaded with Tramadol Fiber Scaffold Characterization

The morphology and structure of the fibers were examined using a scanning electron microscope (SEM) (JSM-7600F, JEOL Ltd., Peabody, MA, USA). Random samples from each group were mounted on cylindrical aluminum stubs and sputter-coated with a 20 nm gold layer for 180 s using a Denton Vacuum Desk V coater (Denton Vacuum, LLC—USA, Moorestown, NJ, USA) Three samples per group were analyzed at magnifications of 50×, 2000×, and 5000× under an accelerating voltage of 10 kV. Images acquired at 5000× were processed with ImageJ software (Version 1.54g) to determine individual fiber diameters. The diameters were tabulated, and fiber size distributions were calculated.

The chemical structure of the pure compounds and synthesized membranes was assessed using a Fourier transform infrared spectrophotometer (Nicolet iS50, ThermoFisher, Madison, WI, USA) over a spectral range of 400–4000 cm^−1^. Three samples of pure PLA, pure TMD, control membranes, and PLA-TMD (80:1) membranes were analyzed. All tests were performed in triplicate to ensure reproducibility. Spectra were processed using OMNIC Spectra 32 software. The data were corrected and linearized for consistent identification of key spectral signals.

Thermal degradation analysis was conducted using a TGA-Q500 equipment (TA Instruments, New Castle, DE, USA). Platinum baskets were tared prior to automatically weighing 4–6 mg of each sample. Samples were heated from 25 °C to 800 °C at a rate of 10 °C/min under a nitrogen atmosphere. Data were processed using Universal Analysis Software (V4.5A, TA Instruments) to identify the onset temperature (To), inflection point (Tp), and temperature of maximum mass loss (Tmax). DSC analysis was performed using a equipment (TA Instruments, New Castle, DE, USA). Approximately 2 mg of each sample was analyzed by heating from 25 °C to 250 °C at a rate of 10 °C/min. The glass transition temperature (Tg), and melting point (Tm) were calculated from the resulting thermograms.

UV-Vis spectroscopy was employed to confirm the presence of TMD in the membranes and its release from the nanofibrous scaffolds. The system was calibrated using a tramadol standard curve at 271 nm. Membrane samples were placed in a Transwell system containing 12 mm diameter inserts with polycarbonate membranes (0.4 µm pores). Each well was filled with 1.5 mL of distilled water and incubated at 37 °C. Aliquots of 1.0 mL were collected at 24, 48, and 72 h for spectroscopic analysis. All experiments were conducted in triplicate. The total amount of drug recovered was quantified, and the percentage release relative to the initial TMD load was determined.

### 4.3. In Vitro Studies

Human fetal osteoblast cells (hFOB 1.19; ATCC CRL-11372) were used to evaluate the biocompatibility of PLA fiber mats and PLA loaded with tramadol. hFOB cells were cultured in 75 cm^2^ flasks containing a 1:1 mixture of Ham’s F12 and Dulbecco’s Modified Eagle Medium (DMEM; Sigma-Aldrich, St. Louis, MO, USA), supplemented with 10% fetal bovine serum (FBS; Gibco, Thermo Fisher Scientific, Waltham, MA, USA), 2.5 mM L-glutamine, and an antibiotic solution (streptomycin 100 μg/mL and penicillin 100 U/mL; Sigma-Aldrich). Cells were incubated in a humidified atmosphere at 37 °C with 5% CO_2_ and 95% air. Cells from passages 2 to 6 were used for all experiments. Prior to biological assays, both PLA and PLA–tramadol membranes were sterilized by UV irradiation for 10 min. To assess cell viability on the scaffolds, hFOB cells were seeded at a density of 1 × 10^4^ cells/mL and evaluated after 24, 48, and 72 h of incubation. Viability was assessed using the resazurin colorimetric assay. At each time point, 20 μL of resazurin solution (BioReagent R7017, Sigma-Aldrich, St. Louis, MO, USA) was added to each sample and incubated for 4 h at 37 °C. Subsequently, 200 μL of supernatant was collected, and absorbance was measured at 545 nm using a ChroMate plate reader (Awareness Technology Inc., Palm City, FL, USA).

The colonization and spreading behavior of hFOB cells seeded onto the PLA scaffolds (1 × 10^4^ cells/mL) were analyzed after 48 h using both scanning electron microscopy (SEM) and epifluorescence microscopy. For fluorescence analysis, hFOB 1.19 cells were first incubated with CellTracker™ Green CMFDA (5-chloromethylfluorescein diacetate; Thermo Fisher Scientific, Waltham, MA, USA) in phenol red-free medium at 37 °C for 30 min. After staining, cells were washed with PBS and incubated for 1 h in complete medium. Cells were then trypsinized, counted, and seeded onto the fiber membranes at the target concentration. After 48 h of incubation, the samples were imaged using an inverted epifluorescence microscope (AE31E, Motic, Xiamen, China). For SEM analysis, after 48 h of culture, the scaffolds were washed three times with PBS, fixed with 2% glutaraldehyde, and dehydrated in a graded ethanol series (25–100%). The samples were air-dried, sputter-coated with gold, and imaged using SEM to evaluate surface morphology and cell attachment.

### 4.4. Statistic Analysis

Fiber’s diameter was analyzed, and average data was compared with T-test. For the biocompatibility tests, the statistics were performed using Analysis of Variance (ANOVA) and the Wilcoxon test for the DSC data analysis.

## Figures and Tables

**Figure 1 ijms-26-06018-f001:**
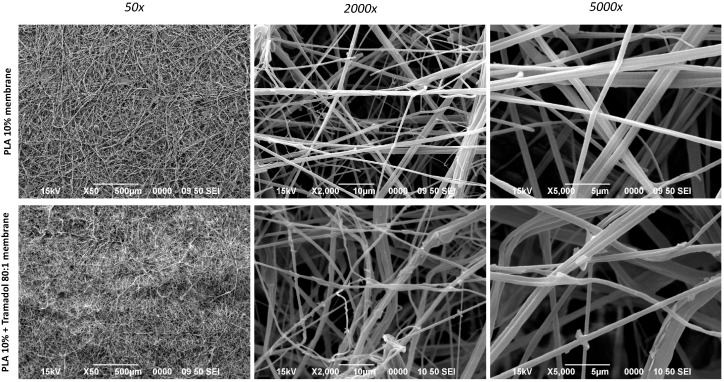
SEM analysis at 50×, 2000×, and 5000× magnifications show control and experimental membranes with a random disposition of the fibers.

**Figure 2 ijms-26-06018-f002:**
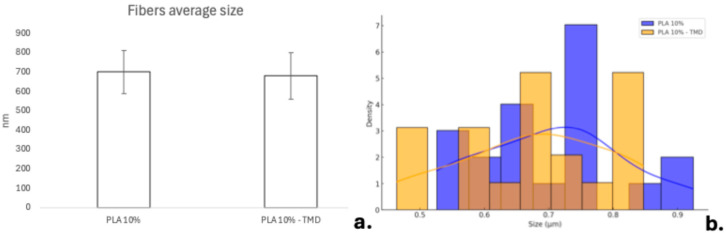
(**a**) Average size of individual fibers for each group. (**b**) Histogram with density curves (fiber size distribution).

**Figure 3 ijms-26-06018-f003:**
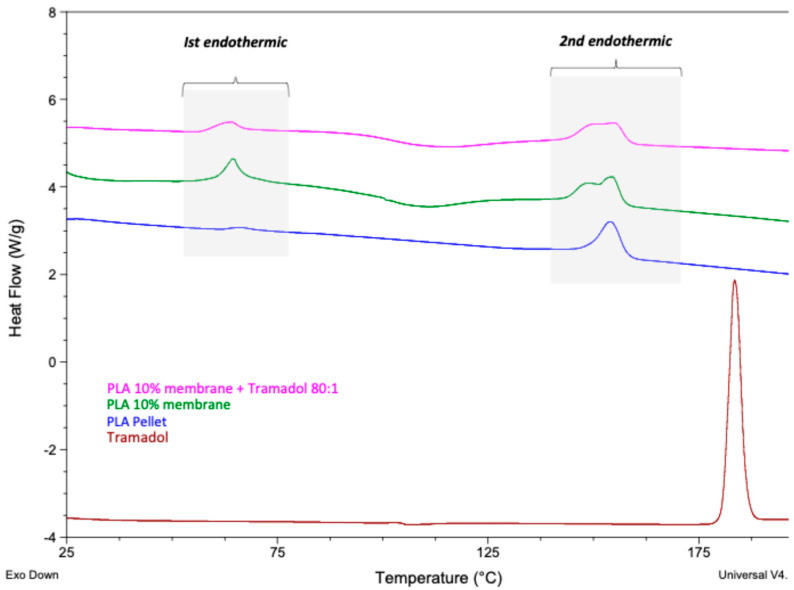
DSC thermograms of neat PLA, 10% PLA membrane, pure TMD, and 10% PLA + TMD membrane.

**Figure 4 ijms-26-06018-f004:**
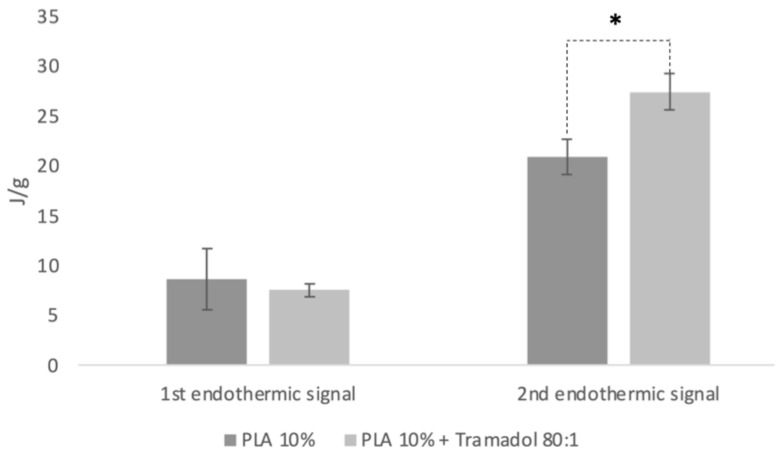
Heat capacity (J/g) of the first and second signal of endothermic capacity for the experimental and control groups. * statistical difference (*p* < 0.05).

**Figure 5 ijms-26-06018-f005:**
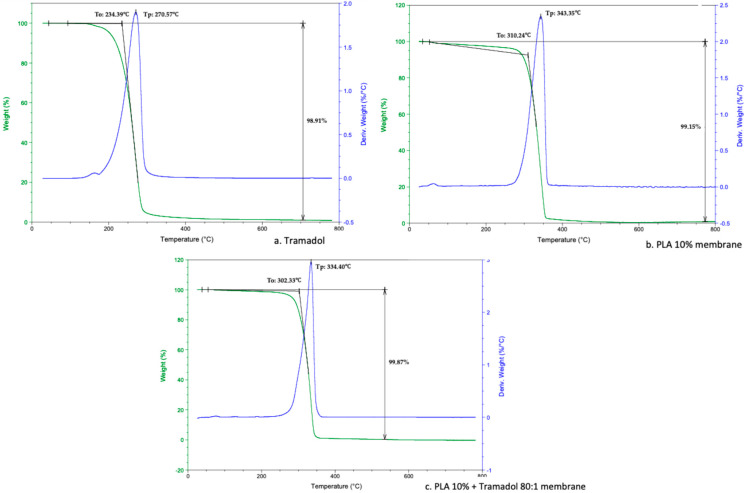
TGA analysis showing onset points (To), inflection point (Tp), and maximum weight loss percentage at 800 °C. (**a**) Pure Tramadol. (**b**) PLA 10% membrane. (**c**) PLA 10% + Tramadol 80:1 membrane.

**Figure 6 ijms-26-06018-f006:**
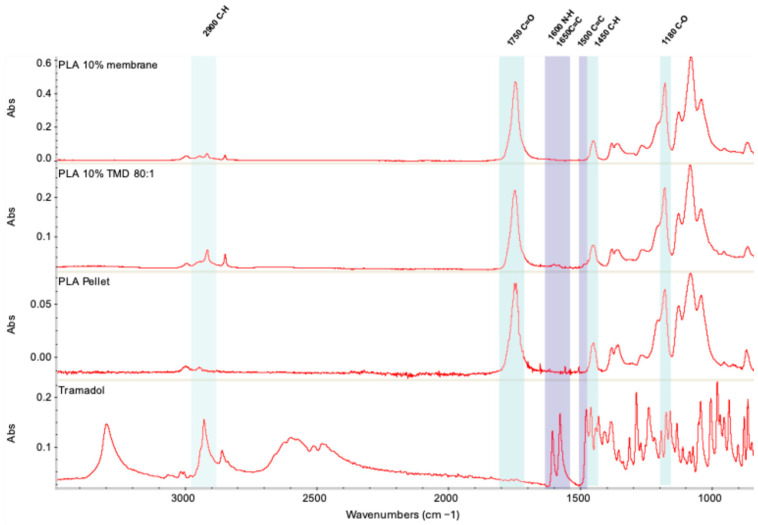
The chemical structure of TMD, pure PLA, PLA 10% membrane, and PLA 10% + TMD membrane were analyzed by FTIR within the range of 400–4000 cm^−1^.

**Figure 7 ijms-26-06018-f007:**
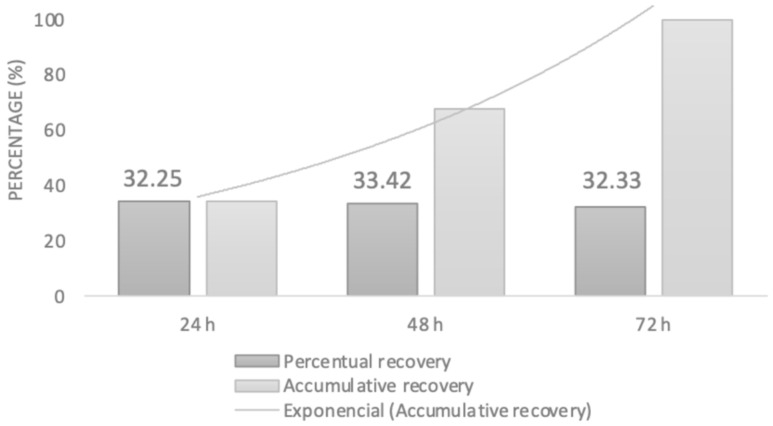
Percentual recovery behavior after hydrolysis experiment for 72 h.

**Figure 8 ijms-26-06018-f008:**
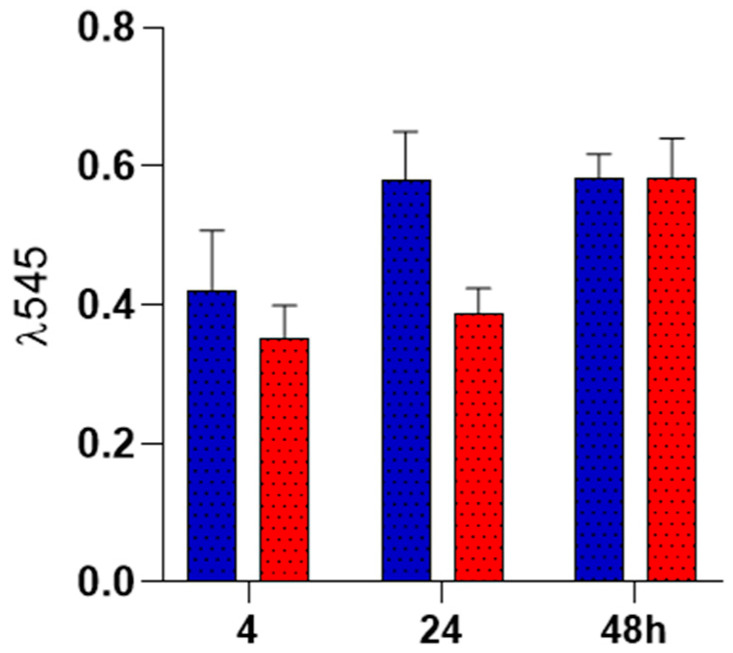
Cell viability response of hFOB after 4, 24, and 48 h cultured onto PLA spun mat loaded with tramadol 80:1 (red) and control 10% PLA spun mat (blue).

**Figure 9 ijms-26-06018-f009:**
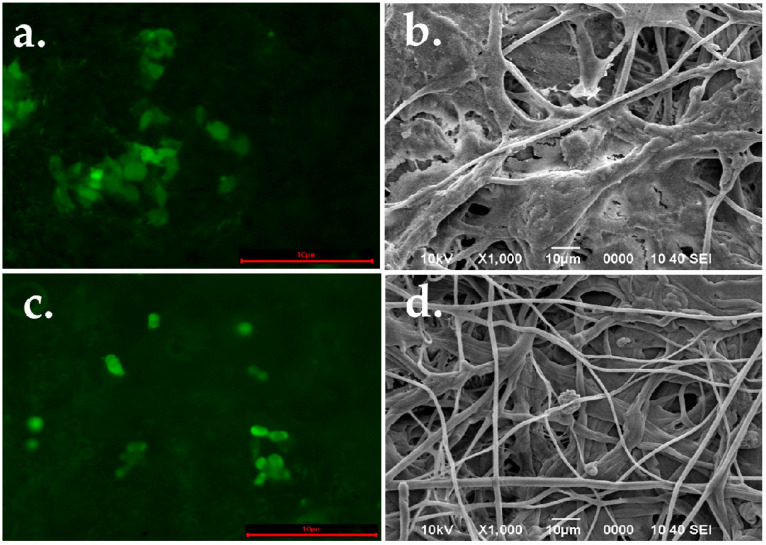
Colonization and spreading pattern of hFOB cultured onto PLA fibers spun mat (**a**,**b**) and PLA loaded with tramadol (**c**,**d**) after 48 h (scale: 10 µm).

**Table 1 ijms-26-06018-t001:** Values of thermogravimetric parameters. TMD decreases the To and Tp of the PLA 10% membranes.

	**Onset Point (To)** **°C**	**Inflection Point (Tp)** **°C**
Tramadol	234.39	270.57
PLA 10% membrane	310.24	343.35
PLA 10% membrane + Tramadol 80:1	302.33	334.40

## Data Availability

Additional data available upon request.

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
