# Peer review of "Characterization of Polylactic Acid Membranes for Local Release of Tramadol"

_ijms, 2025, doi:10.3390/ijms26136018_

Round 1
Reviewer 1 Report
Comments and Suggestions for Authors
In this work, polylactic acid (PLA)-based membranes loaded with tramadol (TMD) in 80:1 ratio, using air jet spinning (AJS) were prepared, characterized by FT-IR and UV-VIS spectroscopies, SEM, DSC, TGA, and biocompatibility assays with human osteoblasts. Authors conclude that PLA membranes loaded with TMD ratio exhibited stable physicochemical characteristics and favorable biocompatibility, supporting their potential use in drug delivery systems.
Technically, the work was well performed. However, several points must be clarified or supported before the manuscript can be reconsidered.
The title: “Design of polylactic acid membranes for controlled release of tramadol,” implies design elements. However, there are no design elements in this work. These elements of design must be included. The title also states the controlled release, but his point was not demonstrated or even planned in the experimental section.
In addition to these major points, several issues must be addressed:
- Authors should indicate the % of TMD recovered after hydrolysis and the expected recovery amount.
- Crystal violet is mentioned in lines 30 and 344 as a staining method; however, it is missing in the experimental section.
- The use of decimal points instead of commas is encouraged.
- Figure 4 should be removed; it does not apport any additional information.
- The significance level at the foot of Figure 5 must be included.
- Figure 6 is unnecessary since data are in Table 1; it should be included as supplementary material.
- The legend of Figure 8 is not consistent with two treatments, PLA 10% and PLA 10%+TMD, and three sampling times.
- L-56 hypersensitivity instead of hypertensive tenderness. L-187 cm-1 instead of cm-1, it must be corrected all along the document.L-189 is symmetric instead of an symmetric. L-200 g instead of G. Results should be appropriately written according to the significant figures of the std. L-258, the mentioned reference “Madeiros et al” should be indicated with the appropriate number as well as in L-271 “Stojanovska et al”.
Many typos and grammar mistakes make the reading difficult.
Author Response
REVIEWER 1
In this work, polylactic acid (PLA)-based membranes loaded with tramadol (TMD) in 80:1 ratio, using air jet spinning (AJS) were prepared, characterized by FT-IR and UV-VIS spectroscopies, SEM, DSC, TGA, and biocompatibility assays with human osteoblasts. Authors conclude that PLA membranes loaded with TMD ratio exhibited stable physicochemical characteristics and favorable biocompatibility, supporting their potential use in drug delivery systems.
Technically, the work was well performed. However, several points must be clarified or supported before the manuscript can be reconsidered.
The title: “Design of polylactic acid membranes for controlled release of tramadol,” implies design elements. However, there are no design elements in this work. These elements of design must be included. The title also states the controlled release, but his point was not demonstrated or even planned in the experimental section.
Thank you for your observation. The title was modified as follows:
Characterization of polylactic acid membranes for local release of tramadol
In addition to these major points, several issues must be addressed:
Authors should indicate the % of TMD recovered after hydrolysis and the expected recovery amount.
Thabk you for the observation. The data was corrected and figure 7 now shows the % of TMD recovered.
Crystal violet is mentioned in lines 30 and 344 as a staining method; however, it is missing in the experimental section.
Thank you for the observation. Corrections are marked in green, between lines 401 and 429. The text was added as follows:
Human fetal osteoblast cells (hFOB 1.19; ATCC CRL-11372) were used to evaluate the biocompatibility of PLA fiber mats and PLA loaded with tramadol. hFOB cells were cultured in 75 cm² flasks containing a 1:1 mixture of Ham’s F12 and Dulbecco’s Modified Eagle Medium (DMEM; Sigma-Aldrich, St. Louis, USA), supplemented with 10% fetal bovine serum (FBS; Biosciences, USA), 2.5 mM L-glutamine, and an antibiotic solution (streptomycin 100 μg/mL and penicillin 100 U/mL; Sigma-Aldrich). Cells were incubated in a humidified atmosphere at 37 °C with 5% CO₂ and 95% air. Cells from passages 2 to 6 were used for all experiments. Prior to biological assays, both PLA and PLA–tramadol membranes were sterilized by UV irradiation for 10 minutes. To assess cell viability on the scaffolds, hFOB cells were seeded at a density of 1 × 10⁴ cells/mL and evaluated after 24, 48, and 72 hours of incubation. Viability was assessed using the resazurin colorimetric assay. At each time point, 20 μL of resazurin solution (BioReagent R7017, CAS 62758-13-8) was added to each sample and incubated for 4 hours at 37 °C. Subsequently, 200 μL of supernatant was collected, and absorbance was measured at 545 nm using a ChroMate plate reader (Awareness Technology, MN, USA).
The colonization and spreading behavior of hFOB cells seeded onto the PLA scaffolds (1 × 10⁴ cells/mL) were analyzed after 48 hours using both scanning electron microscopy (SEM) and epifluorescence microscopy. For fluorescence analysis, hFOB 1.19 cells were first incubated with CellTracker™ Green CMFDA (5-chloromethylfluorescein diacetate) in phenol red-free medium at 37 °C for 30 minutes. After staining, cells were washed with PBS and incubated for 1 hour in complete medium. Cells were then trypsinized, counted, and seeded onto the fiber membranes at the target concentration. After 48 hours of in-cubation, the samples were imaged using an inverted epifluorescence microscope (AE31E, MOTIC). For SEM analysis, after 48 hours of culture, the scaffolds were washed three times with PBS, fixed with 2% glutaraldehyde, and dehydrated in a graded ethanol series (25–100%). The samples were air-dried, sputter-coated with gold, and imaged using SEM to evaluate surface morphology and cell attachment.
The use of decimal points instead of commas is encouraged.
Thank you for the observation. Decimal points were corrected within the text.
Figure 4 should be removed; it does not apport any additional information.
Figure 4 was deleted, and the following figures order was updated.
The significance level at the foot of Figure 5 must be included.
Significance level was included at figure 5. (Line 157 – marked in green).
Figure 6 is unnecessary since data are in Table 1; it should be included as supplementary material.
Thank you for your observation. Although we agree that data is presented in table 1, we include the curves to show the abscence of secundary changes indicating the presence of contamination or the presence of uncharged drug remanents.
The legend of Figure 8 is not consistent with two treatments, PLA 10% and PLA 10%+TMD, and three sampling times.
Thank you for your observation. Figure 8 was deleted.
L-56 hypersensitivity instead of hypertensive tenderness. L-187 cm-1 instead of cm-1, it must be corrected all along the document.L-189 is symmetric instead of an symmetric. L-200 g instead of G. Results should be appropriately written according to the significant figures of the std. L-258, the mentioned reference “Madeiros et al” should be indicated with the appropriate number as well as in L-271 “Stojanovska et al”.
The authors thank so much the reviewer comments. All corrections were included within the manuscript.
Comments on the Quality of English Language
Many typos and grammar mistakes make the reading difficult.
Grammar and writing was doubled checked.
Reviewer 2 Report
Comments and Suggestions for Authors
Reviewer feedback
The research presents a proof of concept to design and deliver biodegradable PLA based membranes for controlled release of tramadol. Below are some comments for authors’ consideration to improve the scientific rigor, accuracy, and quality of the manuscript.
- The abstract needs to include a brief rationale for current work and explicitly highlight the novelty of this research.
- In section 2.1, do the authors refer to the length or thickness of the fibers when specifying the average fiber size?
- How is the density of various fiber sizes determined using SEM analysis?
- Authors have reported decimals using a comma instead of a period. For example, average sizes are reported to be 0,79 um instead of 0.79 um. This inconsistency is observed throughout the manuscript while reporting any numbers in decimals. Please rectify this throughout the text as well as in the tables.
- Moreover, units “uG” should be written as “ug” throughout the manuscript.
- The quality of some of the figures need to be improved to accommodate the standards of this journal. Especially, recommend replotting figures 4, 5, and 8.
- Several major FTIR peaks observed in tramadol do not appear in PLA membrane. This could also indicate some interaction between drug and PLA. Have the authors ruled out any possibility of covalent or ionic interactions between drug and polymer?
- Authors have conducted drug release sampling at 24 h, 48 h and 72 h. However, drug release after 24 h is pretty plateaued and remains constant thereafter. How do the authors justify controlled release to be achieved from this optimized formulation?
- In addition, Fig. 8 displays drug release in ug/mL. Can the authors plot cumulative drug release (% of total drug) over time to get a better understanding of drug release behavior?
- How have the authors differentiated absorbance of degradation products of tramadol having same absorbance wavelength as pure drug during the in vitro release studies and uv-vis spectrometry?
- Are results in Fig. 9 statistically different? The cell viability response at 48 h seem to be very variable compared to 24 h and 48 h. Can the authors explain this behavior?
- Why was the ratio of chloroform/ ethanol kept to 3:1? Can the authors speak to the impact of this ratio on performance of PLA membranes?
- The procedure section needs to be rewritten to include the details- a) include details on how SEM sample preparation was performed; b) what are the SEM measurement parameters on the instrument; c) sample preparation and measurement parameters of FTIR need to be mentioned.
- Why is the in vitro release conducted in water instead of a buffer or biorelevant fluid? Any rationale to this?
Quality of language can be drastically improved. Inconsistencies and inaccuracies in denoting decimals by "'" instead of "."; units such as uG/mL should be written as "ug/mL" according to standard guidelines.
Author Response
REVIEWER 2
Comments and Suggestions for Authors
Reviewer feedback
The research presents a proof of concept to design and deliver biodegradable PLA based membranes for controlled release of tramadol. Below are some comments for authors’ consideration to improve the scientific rigor, accuracy, and quality of the manuscript.
The abstract needs to include a brief rationale for current work and explicitly highlight the novelty of this research.
Thank you for your suggestion, the asbtract was corrected as follows, and marked in green between lines 23 and 41).
Background: Given the growing interest in localized drug delivery to reduce systemic side effects, this study aimed to develop polylactic acid (PLA)-based membranes incorporating tramadol (TMD) using air jet spinning (AJS), ensuring stable physicochemical properties and biocompatibility. To our knowledge, this is the first study to explore the incorporation of TMD into PLA membranes using AJS, a solvent-efficient and scalable technique.
Methods: Two groups were fabricated: 10% PLA membranes (control) and 10% PLA membranes loaded with TMD in an 80:1 ratio (experimental). Characterization included scanning electron microscopy (SEM), differential scanning calorimetry (DSC), thermogravimetric analysis (TGA), Fourier-transform infrared spectroscopy (FT-IR), ultraviolet-visible spectroscopy (UV-VIS), and biocompatibility assays with human osteoblasts using resazurin, crystal violet staining, and 5-chloromethylfluorescein diacetate for fluorescence microscopy.
Results: SEM revealed a homogeneous, randomly distributed fiber pattern, with diameters under 5 µm and no structural voids. DSC and TGA indicated that TMD was uniformly incorporated, increased the thermal capacity, and slightly lowered the onset and inflection degradation temperatures. FT-IR confirmed the chemical compatibility of TMD with PLA, showing no structural alterations. UV-VIS detected sustained TMD release over 72 h. Biocompatibility tests showed no cytotoxic effects; cell viability and proliferation in TMD-loaded membranes were comparable to controls. Statistical analysis used ANOVA and Wilcoxon tests.
Conclusions: 10% PLA membranes loaded with TMD at an 80:1 ratio exhibited stable physicochemical characteristics and favorable biocompatibility. These findings support the innovative potential of AJS-fabricated PLA membranes as a promising platform for localized analgesic delivery.
In section 2.1, do the authors refer to the length or thickness of the fibers when specifying the average fiber size?
Thank you for the observation. We refered to thicknes. The correction was marked in green, and can be read between lines 362 and 369. The text was complemented as follows:
The morphology and structure of the fibers were examined using a scanning electron microscope (SEM, JSM-7600F, JEOL, USA). Random samples from each group were mounted on cylindrical aluminum stubs and sputter-coated with a 20 nm gold layer for 180 seconds using a Denton Vacuum Desk V coater. Three samples per group were analyzed at magnifications of 50×, 2000×, and 5000× under an accelerating voltage of 10 kV. Images acquired at 5000× were processed with ImageJ software (Version 1.54g) to determine in-dividual fiber diameters. The diameters were tabulated, and fiber size distributions were calculated.
How is the density of various fiber sizes determined using SEM analysis?
SEM images at 5000x were analized using Image J software, and individual fibers thickness was recorded and analyzed. The data was used to obtain the average fibers thickness and its distribution.
Authors have reported decimals using a comma instead of a period. For example, average sizes are reported to be 0,79 um instead of 0.79 um. This inconsistency is observed throughout the manuscript while reporting any numbers in decimals. Please rectify this throughout the text as well as in the tables.
The writing inconsistency was corrected.
Moreover, units “uG” should be written as “ug” throughout the manuscript.
The grams unit was corrected.
The quality of some of the figures need to be improved to accommodate the standards of this journal. Especially, recommend replotting figures 4, 5, and 8.
The quality of all figures was improved.
Several major FTIR peaks observed in tramadol do not appear in PLA membrane. This could also indicate some interaction between drug and PLA. Have the authors ruled out any possibility of covalent or ionic interactions between drug and polymer?
Thank you for this important observation. Although all the characteristic signals of tramadol were clearly identifiable in the pure drug, these peaks were scarcely detectable in the drug-loaded membrane. This can be attributed to the limited penetration depth of the FT-IR laser beam, which may reach only ~0.5 μm. Additionally, the low drug-to-polymer ratio (80:1) and the homogeneous distribution of tramadol within the polymeric matrix may result in an FT-IR spectrum where tramadol-specific peaks are not readily distinguishable. Furthermore, potential interactions between tramadol and the polymer could modify the drug's chemical environment, thereby affecting its infrared absorption profile. To confirm the presence of tramadol, exclude the possibility of chemical modification, and evaluate its release capacity, a drug release assay was conducted. The successful recovery of tramadol from the membranes confirmed its presence and availability within the polymeric system.
Authors have conducted drug release sampling at 24 h, 48 h and 72 h. However, drug release after 24 h is pretty plateaued and remains constant thereafter. How do the authors justify controlled release to be achieved from this optimized formulation?
The objective of the recovery experiment was not to evaluate the release profile, but to confirm the presence and recovery posibility from the membranes. The presence of drug was observed in every sampling period, with a trent to decrease over time.
In addition, Fig. 8 displays drug release in ug/mL. Can the authors plot cumulative drug release (% of total drug) over time to get a better understanding of drug release behavior?
Yes, the image was corrected to improve the understanding of this experiment, and to clarify the intention fo the release assay.
How have the authors differentiated absorbance of degradation products of tramadol having same absorbance wavelength as pure drug during the in vitro release studies and uv-vis spectrometry?
The authors thank the reviewer for the observation. The experiment was not conducted to analyze if the recovery of the drug corresponds to pure Tramadol or its degradation products. In this case, the absorbance at 271nm wavelenght was analyzed to confirm the presence of tramadol, since the polymer signal is identifiable at 250nm.
Are results in Fig. 9 statistically different? The cell viability response at 48 h seem to be very variable compared to 24 h and 48 h. Can the authors explain this behavior?
By the end of the experiment, no statistically significant differences were observed between the two groups. Although a slight difference in absorbance was detectable at 24 hours, this difference was no longer evident after 48 hours, indicating that cellular behavior had equilibrated and was comparable between the control and experimental groups. It is important to emphasize that once stabilization occurred, both groups maintained a similar biological response, even while the drug continued to be released.
Why was the ratio of chloroform/ ethanol kept to 3:1? Can the authors speak to the impact of this ratio on performance of PLA membranes?
The chloroform/ethanol combination is the regular solution created to prepare the PLA before the synthesis. This was confirmed in previous works, and this solution is not related to an increase citotoxicity. Here are an important previous works that evaluated the same methodology
Mena-Porras E, Contreras-Aleman A, Guevara-Hidalgo MF, Avendaño Soto E, Batista Menezes D, Alvarez-Perez MA, Chavarría-Bolaños D. Comparison of Two Synthesis Methods for 3D PLA-Ibuprofen Nanofibrillar Scaffolds. Pharmaceutics. 2025 Jan 14;17(1):106.
The procedure section needs to be rewritten to include the details- a) include details on how SEM sample preparation was performed; b) what are the SEM measurement parameters on the instrument; c) sample preparation and measurement parameters of FTIR need to be mentioned.
The SEM and FT-ir methods were rewritten, and can be found marked in green between lines 362 and 377; and can be read as follows:
The morphology and structure of the fibers were examined using a scanning electron microscope (SEM, JSM-7600F, JEOL, USA). Random samples from each group were mounted on cylindrical aluminum stubs and sputter-coated with a 20 nm gold layer for 180 seconds using a Denton Vacuum Desk V coater. Three samples per group were analyzed at magnifications of 50×, 2000×, and 5000× under an accelerating voltage of 10 kV. Images acquired at 5000× were processed with ImageJ software (Version 1.54g) to determine in-dividual fiber diameters. The diameters were tabulated, and fiber size distributions were calculated.
The chemical structure of the pure compounds and synthesized membranes was assessed using a Fourier transform infrared spectrophotometer (Nicolet iS50, Ther-moFisher, Madison, USA) over a spectral range of 400–4000 cm⁻¹. Three samples of pure PLA, pure TMD, control membranes, and PLA-TMD (80:1) membranes were analyzed. All tests were performed in triplicate to ensure reproducibility. Spectra were processed using OMNIC Spectra 32 software. The data were corrected and linearized for consistent identification of key spectral signals.
Why is the in vitro release conducted in water instead of a buffer or biorelevant fluid? Any rationale to this?
Since tramadol is a hydrophilic drug, and as a departure point we only included one inert medium to evaluate the recovery of the drug once PLA was hydrolized. However, we recognize the importante to repeat this experiment in a different medium as well. This was written in the limitations section at the endo of the discussion.
Comments on the Quality of English Language
Quality of language can be drastically improved. Inconsistencies and inaccuracies in denoting decimals by "'" instead of "."; units such as uG/mL should be written as "ug/mL" according to standard guidelines.

Round 2
Reviewer 1 Report
Comments and Suggestions for Authors
Authors improved their document. However, they carelessly included a new figure numbered as Figure 7.
Figure 7 and the footnote should be improved: replace commas with periods, the end of the exponential line is not visible, separate values from units, and replace 'recovey' with 'recovery'.
Author Response
Thank you very much for your observations. A new version of figure 7 was included, and it was correctly cited within the text.
Reviewer 2 Report
Comments and Suggestions for Authors
Authors have satisfactorily addressed the comments and improved on the manuscript.
Author Response
Thank you very much for your comments.
Round 3
Reviewer 1 Report
Comments and Suggestions for Authors
In this work, polylactic acid (PLA)-based membranes loaded with tramadol (TMD) in 80:1 ratio, using air jet spinning (AJS) were prepared, characterized by FT-IR and UV-VIS spectroscopies, SEM, DSC, TGA, and biocompatibility assays with human osteoblasts. Authors conclude that PLA membranes loaded with TMD ratio exhibited stable physicochemical characteristics and favorable biocompatibility, supporting their potential use in drug delivery systems.
The authors have addressed the initial issues, and I agree that this manuscript should be published in its current form.